# Characterization of TelE, a T7SS LXG Effector Exhibiting a Conserved C-Terminal Glycine Zipper Motif Required for Toxicity

Wooi Keong Teh,[a] Yichen Ding,[a*] Francesca Gubellini,[b] Alain Filloux,[a,c] Claire Poyart,[d] Michael Givskov,[a,g] Shaynoor Dramsi[e,f]

[a]Singapore Centre for Environmental Life Sciences Engineering, Nanyang Technological University, Singapore, Singapore

[b]Institut Pasteur, Unité de Microbiologie Structurale, Paris, France

[c]Centre for Bacterial Resistance Biology, Department of Life Sciences, Imperial College London, London, United Kingdom

[d]Université de Paris, Assistance Publique Hôpitaux de Paris, Service de Bactériologie, Centre National de Référence des Streptocoques, Groupe Hospitalier Paris Centre site Cochin, Paris, France

[e]Institut Pasteur, Université Paris Cité, CNRS UMR6047, Biology of Gram-positive Pathogens Unit, Paris, France

[f]Centre National de la Recherche Scientifique (CNRS) UMR2001, Paris, France

[g]Costerton Biofilm Centre, Department of Immunology and Microbiology, University of Copenhagen, Denmark

**ABSTRACT** *Streptococcus gallolyticus* subsp. *gallolyticus* (*SGG*) is an opportunistic bacterial pathogen strongly associated with colorectal cancer. Here, through comparative genomics analysis, we demonstrated that the genetic locus encoding the type VIIb secretion system (T7SSb) machinery is uniquely present in *SGG* in two different arrangements. *SGG* UCN34 carrying the most prevalent T7SSb genetic arrangement was chosen as the reference strain. To identify the effectors secreted by this secretion system, we inactivated the *essC* gene encoding the motor of this machinery. A comparison of the proteins secreted by UCN34 wild type and its isogenic Δ*essC* mutant revealed six T7SSb effector proteins, including the expected WXG effector EsxA and three LXG-containing proteins. In this work, we characterized an LXG-family toxin named herein TelE promoting the loss of membrane integrity. Seven homologs of TelE harboring a conserved glycine zipper motif at the C terminus were identified in different *SGG* isolates. Scanning mutagenesis of this motif showed that the glycine residue at position 470 was crucial for TelE membrane destabilization activity. TelE activity was antagonized by a small protein TipE belonging to the DUF5085 family. Overall, we report herein a unique *SGG* T7SSb effector exhibiting a toxic activity against nonimmune bacteria.

**IMPORTANCE** In this study, 38 clinical isolates of *Streptococcus gallolyticus* subsp. *gallolyticus* (*SGG*) were sequenced and a genetic locus encoding the type VIIb secretion system (T7SSb) was found conserved and absent from 16 genomes of the closely related *S. gallolyticus* subsp. *pasteurianus* (*SGP*). The T7SSb is a bona fide pathogenicity island. Here, we report that the model organism *SGG* strain UCN34 secretes six T7SSb effectors. One of the six effectors named TelE displayed a strong toxicity when overexpressed in *Escherichia coli*. Our results indicate that TelE is probably a pore-forming toxin whose activity can be antagonized by a specific immunity protein named TipE. Overall, we report a unique toxin-immunity protein pair and our data expand the range of effectors secreted through T7SSb.

**KEYWORDS** *Streptococcus gallolyticus*, type VIIb secretion system, pore-forming toxin, LXG family toxin, glycine zipper, LXG effector, T7SSb

**S**treptococcus gallolyticus subsp. *gallolyticus* (*SGG*) is a commensal bacterium in the rumen of herbivores. The first complete genome of strain UCN34 provided clear insights for its adaptation to the rumen with the discovery of a high number of enzymes

Address correspondence to Shaynoor Dramsi, shaynoor.dramsi@pasteur.fr, or Michael Givskov, mgivskov@sund.ku.dk.

*Present address: Yichen Ding, National Public Health Laboratory, Singapore, Singapore.

The authors declare no conflict of interest.

involved in complex carbohydrates degradation and in tannin detoxification (1). *SGG* is also an opportunistic pathogen causing septicemia and endocarditis in the elderly who often have concomitant colon tumors (2–9). *SGG* was formerly known as *S. bovis* biotype I belonging to the large *S. bovis/equinus* complex, which is comprised of several opportunistic pathogens and nonpathogenic bacteria. In this complex, *SGG* is assigned into a clade with two other closely related subspecies named *S. gallolyticus* subsp. *pasteurianus* (referred to here as *SGP*) and *S. gallolyticus* subsp. *macedonicus* (referred to here as *SGM*) (10, 11). *SGP* can cause neonatal meningitis (12, 13) whereas *SGM* is generally recognized as a safe nonpathogenic strain (14, 15).

The type VIIb secretion system (T7SSb) is a specialized secretion system exploited by *Firmicutes* to secrete various substrates or effectors involved in host immune system modulation or bacterial competition (16–20). T7SSb is distantly related to the type VII secretion system originally discovered in *Mycobacterium tuberculosis*. These two evolutionarily distant but related secretion systems exhibit two components only in common; a membrane-bound ATPase (FtsK/SpoIII, EssC/EccC) constituting the motor of the secretion machinery and a secreted effector named EsxA/ESAT6 belonging to the WXG100 family (21). In *Firmicutes*, the T7SSb core machinery is composed of four additional proteins, namely, EssA, EsaA, EsaB, and EssB. This specialized secretion system is involved in the secretion of a variety of effectors including (i) WXG100 proteins such as EsxA and EsxB, which are short alpha-helical proteins with a characteristic WXG motif in the middle of the sequence (22); (ii) proteins of approximately 100 amino acids without a WXG motif (named WXG100-like proteins) such as EsxC and EsxD (23, 24); and (iii) polymorphic toxins containing an N-terminal LXG domain and a variable C-terminal domain displaying antibacterial activity or implicated in virulence such as TelA, TelB, TelC, TelD, TspA, and EsxX (18, 25–28). The LXG domain is predicted to adopt an alpha-helical structure that is reminiscent of the WXG100 proteins. Previous studies revealed that these LXG-containing effectors function as a NADase toxin (TelB) (18), a lipid II phosphatase (TelC) (18), nucleases (EsaD, YobL, and YxiD) (16, 27, 29), or a membrane depolarizing toxin (TspA) (25).

Importantly, the T7SSb of *SGG* strain TX20005 was shown to contribute to the development of colon tumors in a murine model (30). Here, using comparative genomics, we identified that the T7SSb locus residing in a 26-kb genomic island is uniquely present in *SGG* but neither in *SGP* nor in *SGM*. Subsequently, we found that six proteins were secreted through the T7SSb machinery of the reference strain UCN34, including the WXG100 effector EsxA, two hypothetical proteins, and three LXG family proteins, two of them sharing similarities to TelC, a lipid II phosphatase characterized in *Streptococcus intermedius* (18). One of the unknown LXG family proteins was characterized herein and named TelE. We show that TelE is a membrane-destabilizing protein whose activity is antagonized by a specific immunity protein TipE.

## RESULTS

**A T7SSb locus specific to *SGG*.** To identify the specific pathogenic trait(s) of *SGG*, comparative genomics was performed on newly sequenced 13 *SGP* and 38 *SGG* clinical isolates obtained from the National Reference Center of Streptococci in France. Phylogenetic analysis of these newly sequenced genomes, together with the complete genomes of *SGG* and *SGP* publicly available on NCBI GenBank, assigned *SGG* and *SGP* into two distinct clades, confirming their correct classification (Fig. S1a). BLASTn-based comparative genomic analysis of the reference genome UCN34 together with the representative *SGG* and *SGP* genomes uncovered a genomic island unique to *SGG* (Fig. 1A; Fig. S1b). This genomic island contains the six genes (*esxA*, *essA*, *esaB*, *essB*, *essC,* and *esaA*) encoding the core components of the type VIIb secretion system (T7SSb) machinery (Fig. 1B; Fig. S1b).

An extended protein homology search of each T7SSb core component across all the newly sequenced genomes confirmed the conservation of T7SSb in *SGG* genomes and its absence in the phylogenetically closest relatives *SGP* and *SGM* (Fig. S1b). A

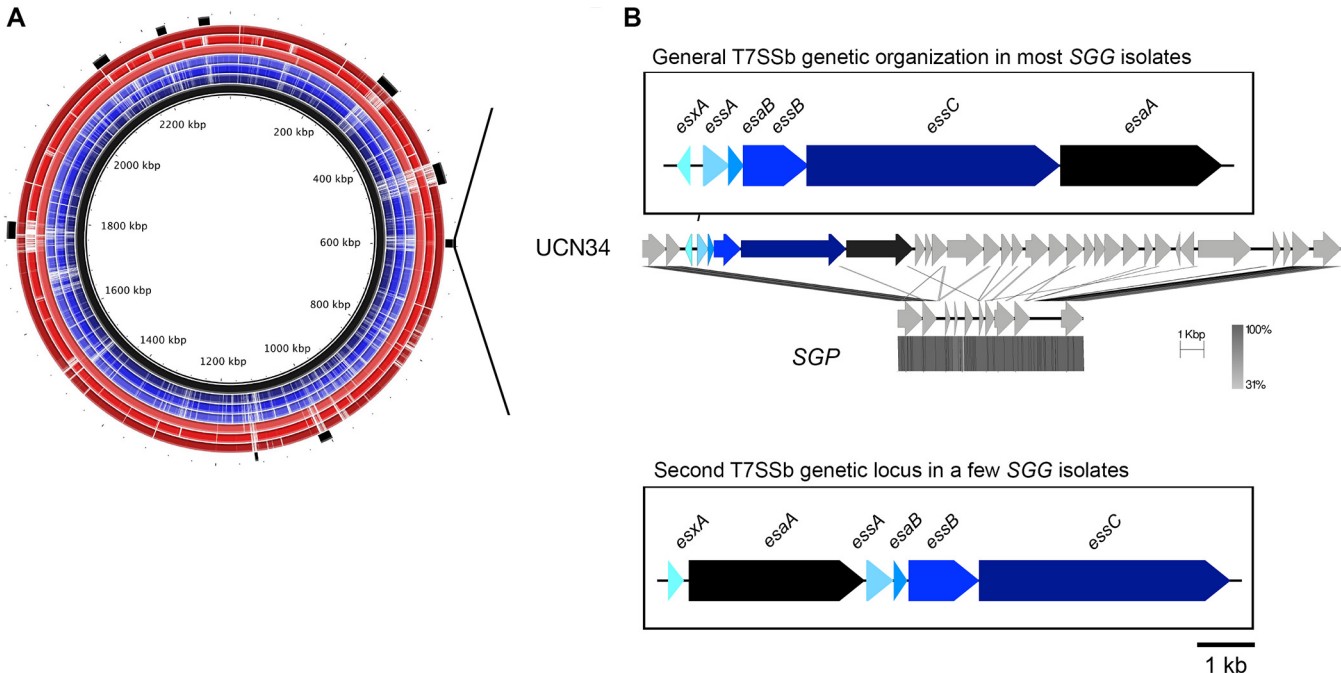

**FIG 1** A T7SSb locus specific to *SGG*. (A) An overview of the BLASTn-based comparative genomics analysis. From the innermost circle, the reference genome *SGG* UCN34 (black ring); followed by *SGP* isolates (blue rings, Gallo1, Gallo4, and Gallo6); and *SGG* isolates (red rings, Gallo37, Gallo43, and Gallo51). The outermost black blocks represent the genomic islands in UCN34 genome predicted by IslandViewer 4. White spaces within each ring represent ≤50% similarities of the respective genomic regions to UCN34. (B) Genetic organization of the two types of T7SSb found in *SGG* clinical isolates and absent in *SGP*. Each T7SSb gene encoding the secretion machinery was color coded and labeled accordingly.

closer inspection of the T7SSb genetic locus uncovered two different gene arrangements in the *SGG* isolates sequenced so far. In a vast majority of *SGG* isolates, the T7SSb locus is arranged in the order of *esxA/essA-esaB-essB-essC-esaA*, as in our reference strain UCN34. Notably, *esxA* is transcribed in the opposite direction from the other five genes *essA-esaB-essB-essC-esaA*, a unique feature distinct from other *Firmicutes*. In contrast, in a small minority of *SGG* isolates including the U.S. isolate TX20005, the T7SSb genes are arranged in a different order of *esxA-esaA-essA-esaB-essB-essC* with an evident shuffling of *esaA*, and the six genes transcribed in the same direction (Fig. 1B; Fig. S1b). A detailed comparison of the genetic organization of the entire T7SSb locus in the representative *SGG* strains UCN34 and TX20005 is shown in Fig. S2. In addition to a different gene organization of the core machinery, the set of putative T7SSb effectors is different and appears to be strain specific.

**SGG UCN34 secretes six T7SSb effectors.** To identify the T7SSb effectors, we inactivated the T7SSb machinery, through in-frame deletion of *essC* gene encoding the main ATPase, in the strain UCN34. We next compared the secretome of the UCN34 wild-type (WT) strain to that of the Δ*essC* mutant. This approach identified six proteins whose abundance was significantly different in the supernatant of the two strains (Fig. 2A; Table S2). This includes the ubiquitous WXG100 effector (EsxA/Gallo_0553) whose level was about 38-fold less abundant in the Δ*essC* mutant, confirming that the Δ*essC* mutant is a bona fide T7SSb-deficient strain. In addition, five proteins, namely, Gallo_0559, Gallo_0560, Gallo_0562, Gallo_1068, and Gallo_1574, were detected only in the secretome of the UCN34 WT but not in the Δ*essC* mutant.

Prediction of the putative function of these effectors by *in silico* SMART analysis revealed that Gallo_0559 contains a domain of unknown function DUF5082, Gallo_0560 as a TIGR04197 family protein, whereas Gallo_0562, Gallo_1068, and Gallo_1574 contain the prototypical N-terminal LXG domain (pfam04740) found in other T7SSb effectors (Fig. 2B). Protein homology search indicated that Gallo_1068 and Gallo_1574 are homologous (>98% coverage; >40% similarity) to the *Streptococcus intermedius* TelC effector that functions as a lipid II phosphatase (18). Similar to TelC, both Gallo_1068 and

**A**

| Locus tag | Total peptide count | | Coverage | Fold change* | Adjusted p-value# |
|---|---|---|---|---|---|
| | UCN34 WT | ΔessC | | | |
| Gallo_0553 | 401.67 ± 41.50 | 10.67 ± 3.06 | 100.00% | 37.64 | 6.71 x 10⁻¹⁵ |
| Gallo_0559 | 34.00 ± 7.00 | N.D. | 47.83% | N.A. | 6.71 x 10⁻¹⁵ |
| Gallo_0560 | 8.33 ± 0.58 | N.D. | 48.39% | N.A. | 6.71 x 10⁻¹⁵ |
| Gallo_0562 | 42.33 ± 6.04 | N.D. | 29.77% | N.A. | 6.71 x 10⁻¹⁵ |
| Gallo_1068 | 9.33 ± 3.09 | N.D. | 17.50% | N.A. | 2.42 x 10⁻⁸ |
| Gallo_1574 | 13.00 ± 1.00 | N.D. | 20.77% | N.A. | 6.71 x 10⁻¹⁵ |

* value expressed as WT vs ΔessC

**B**

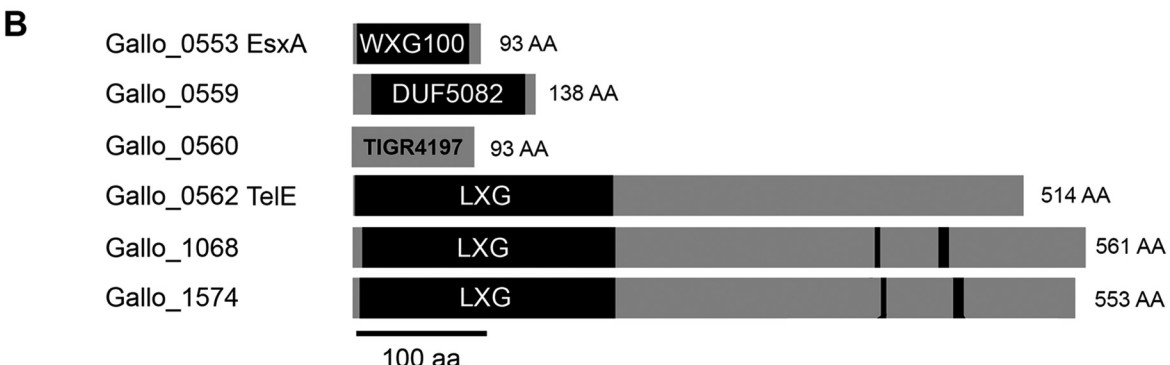

**C**

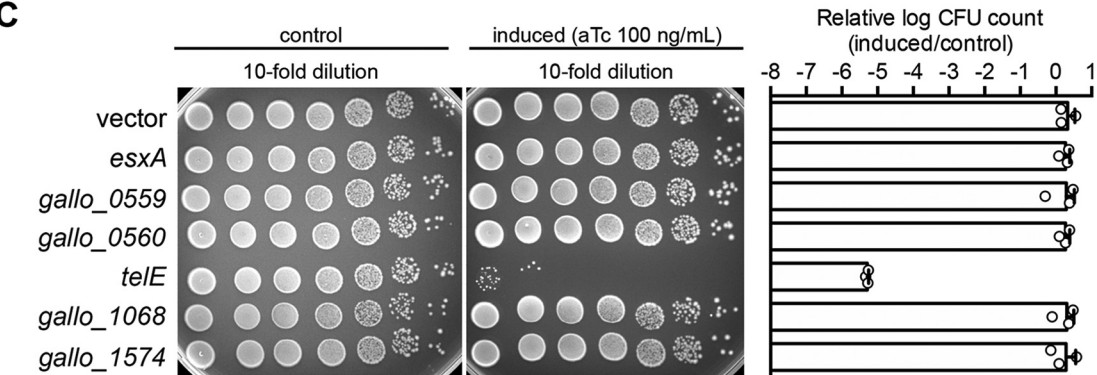

**FIG 2** A *SGG* T7SSb effector exhibits toxicity in *E. coli*. (A) List of proteins that were significantly more abundant in the secretome of the UCN34 wild type than the ΔessC mutant. The total peptide count of each protein was expressed as means ± standard deviation from three biological replicates. N.D., not detected; N.A., not applicable. (B) The domain architecture of the six proteins identified in panel A. Both Gallo_1068 and Gallo_1574 are homologous to the lipid II phosphatase TelC. The conserved aspartate-rich motifs essential for TelC activity are indicated by black boxes. (C) Viability of the bacterial cells harboring either an empty vector (pTCVerm-P$_{tetO}$) or a vector encoding the full-length coding sequences of the candidate genes with the expression inducible by anhydrotetracycline (aTc). Bar chart on the right shows the logarithm-transformed CFU count of the candidate gene-expressing cells relative to the control. Error bars represent mean ± standard deviation (*n* = 3).

Gallo_1574 contain an intact C-terminal aspartate-rich motif critical for TelC enzymatic activity (Fig. 2B). The third LXG-containing effector (Gallo_0562) is a protein of 514 amino acids that we propose to rename TelE in accordance with the gene nomenclature (Toxin exported by Esx with LXG domain) proposed for *S. intermedius* (18). Of note, four out of six proteins identified in the secretome of UCN34 WT are encoded in the T7SSb locus and three are encoded by genes likely transcribed as an operon, namely, *gallo_0559*, *gallo_0560*, and *telE* (*gallo_0562*).

**TelE is toxic when overexpressed in *Escherichia coli*.** LXG family proteins are polymorphic toxins implicated in interbacterial competition (18). However, we failed to

observe any role of *SGG* T7SSb in interbacterial competition under standard laboratory conditions. We reasoned that this could be due to the low expression of T7SSb effectors as reflected by the low total peptide counts in the proteomics data (Fig. 2A) and decided to test their potential antibacterial activity through heterologous and inducible expression in *Escherichia coli* DH5α, as reported for other T7SSb effectors such as TspA and TelD (25, 28). As shown in Fig. 2C, overexpression of *telE*, but none of the other T7SSb effectors, strongly compromised *E. coli* viability (Fig. 2C).

TelE is encoded in the vicinity of the T7SSb core machinery locus (Fig. 3A). This prompted us to examine the prevalence and the conservation of TelE among the *SGG* clinical isolates. As shown in Fig. S3, TelE homologs were found in over 90% of *SGG* isolates. Based on their primary amino acid sequences, TelE homologs can be categorized into seven variants named TelE1 (initial TelE found in UCN34) through TelE7. Notably, multiple copies of TelE homologs can be found in a few *SGG* genomes. Some of these TelE homologs are N-terminally truncated, resembling the orphan modules commonly seen in polymorphic toxins (31) (Fig. S3). To assess the antimicrobial activity of these TelE homologs, each TelE variant was expressed in *E. coli*. As shown in Fig. 3B, all the TelE variants exhibited a strong toxicity upon induction in *E. coli*. Interestingly, a multiple sequence alignment of these seven TelE variants uncovered a highly conserved sequence motif at the C terminus (Fig. S4). This motif, rich in small hydrophobic amino acids alternating with conserved glycine residues, is similar to the glycine zipper motif (GxxxGxxxG) found in many membrane proteins (32) (Fig. 3C). Three highly conserved glycine residues were identified at positions 458, 470, and 474, and three less-conserved glycine residues were found at positions 466, 478, and 480 (Fig. 3C and D). Single replacement of each glycine residue to a slightly bulkier valine residue demonstrated that those at positions 466, 470, and 474 were key for TelE toxicity in *E. coli* (Fig. 3E), with the replacement of the glycine residue at position 470 showing the strongest impact on TelE activity (Fig. 3E). Together, these data showed that nearly all the *SGG* isolates sequenced as to date encode a functional TelE homolog whose primary sequence varies from one isolate to another but all seven TelE variants displaying a conserved glycine zipper motif at the C terminus essential for TelE antibacterial activity.

**TelE compromises bacterial membrane integrity.** Structural modeling of TelE indicates a high number of alpha helices and the C-terminal domain of TelE exhibit a small stretch of amino acids weakly homologous to bacteriocin IIc and glycine zipper motif (Fig. 4A). Alpha-helical structures with glycine zipper motifs are the two main characteristics of membrane proteins with a high propensity to form homo-oligomeric ion channels (32), suggesting that TelE might have a pore-forming activity. To investigate this possibility, we used propidium iodide, a large molecule (668 Da) that is conventionally used to assess the presence of membrane pores or membrane disruption in bacterial cells (33, 34). Propidium iodide can only enter the cellular cytoplasmic space through membrane lesions or large membrane pores to subsequently bind to intracellular DNA and emit a red fluorescence signal. Using time-lapse fluorescence microscopy, we observed a gradual increase of red fluorescence signal in *E. coli* cells expressing TelE but not in *E. coli* cells carrying the empty vector (Fig. 4B), indicating that TelE expression resulted in an influx of propidium iodide into the cells. Intriguingly, this influx was preceded immediately by shrinkage of the bacterial cell visible by phase-contrast microscopy (Fig. 4C). Similar morphological changes have been observed on cells experiencing water loss or cytoplasmic content leakage (35, 36), which can be due to the presence of membrane pores. Our microscopy observations suggest that TelE can alter bacterial membrane integrity, likely by forming membrane pores in *E. coli* cells. To monitor TelE expression, a chimeric TelE variant C-terminally fused with superfolder green fluorescent protein (TelE-sfGFP) was constructed and GFP fluorescence signals were measured to quantify TelE expression (Fig. 4D). Of note, TelE-sfGFP retains TelE toxicity in *E. coli* cells (Fig. 4D, right). The nontoxic version of TelE (TelE$_{G470V}$-sfGFP) was constructed in parallel. As shown in Fig. 4D, an increase in GFP fluorescence was seen upon induction of TelE expression over time from 10 to 60 min. The TelE$_{G470V}$-

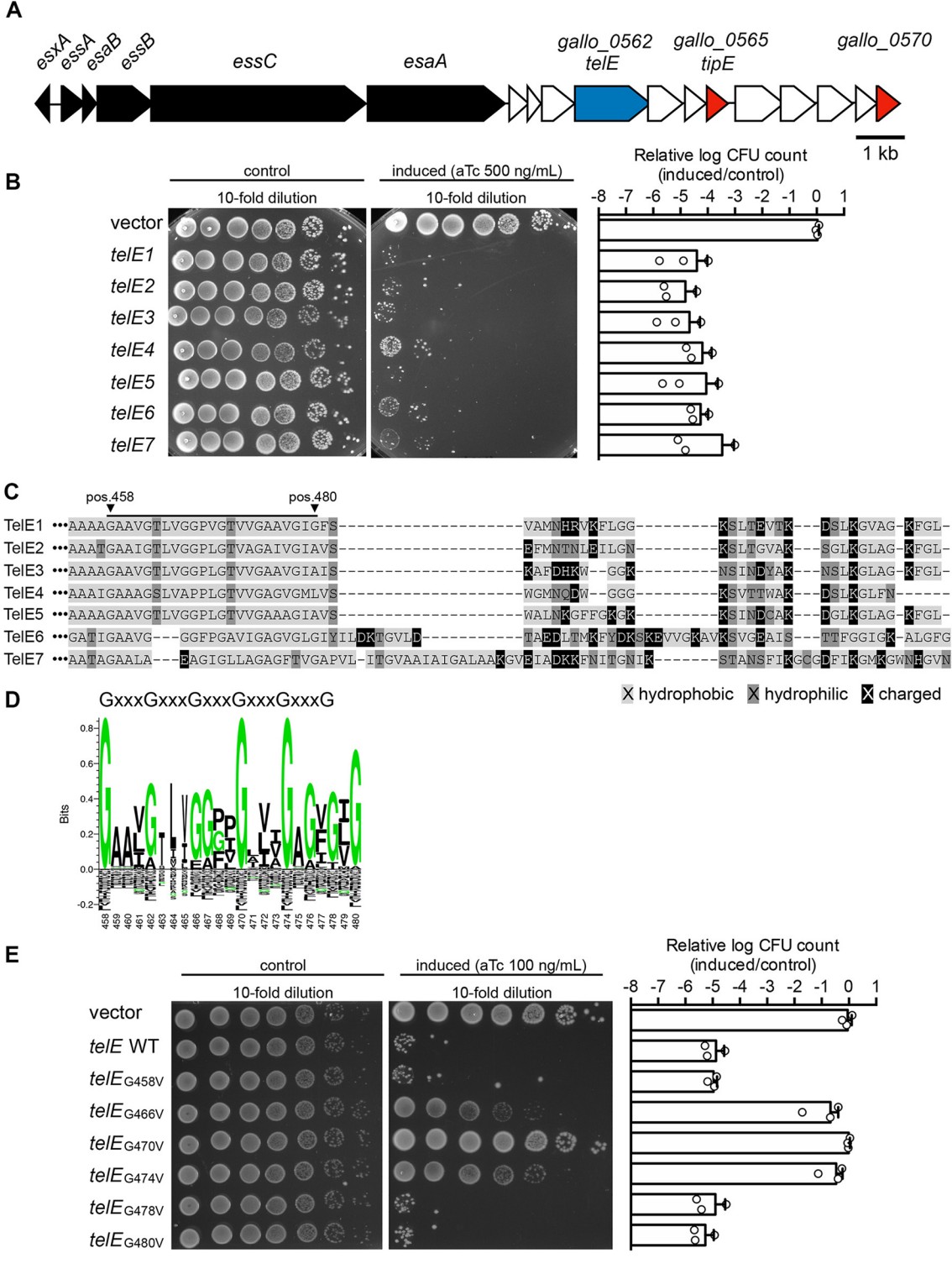

**FIG 3** T7SSb effector TelE contains a C-terminal glycine zipper motif essential for its activity. (A) The genomic region encoding the T7SSb core components TelE and TipE in *SGG* UCN34. Black, T7SSb core components; blue, TelE (LXG domain-containing protein); red, DUF5085 domain-containing protein; white, hypothetical proteins. (B) Viability of the *E. coli* cells harboring either an empty vector (pTCVerm-P$_{tetO}$) or a vector encoding various *telE* variants (*telE1* to *telE7*) with the expression inducible by anhydrotetracycline (aTc). Bar chart on the right shows the logarithm-transformed CFU count of the *telE*-expressing cells relative to the control. Error bars represent mean ± standard deviation (*n* = 3). (C) Alignment of the C terminus sequence of the *telE* variants expressed in panel B revealed a conserved region (underlined) enriched with hydrophobic residues. (D) Sequence logo of the underlined region in panel C generated from all aligned TelE variants identified a sequence motif highly similar to glycine zipper motif (GxxxGxxxG). The number at the bottom of the sequence logo represents the position of the respective amino acid residue in reference to the TelE sequence from UCN34. Shrunk residues (i.e., residues at position 462 to 465) reflected the presence of gaps in the sequence alignment. (E) Viability of the *E. coli* cells harboring either an empty vector (pTCVerm-P$_{tetO}$) or a vector encoding the mutated *telE* with the expression inducible by anhydrotetracycline (aTc). Bar chart on the right shows the logarithm-transformed CFU count of the *telE*-expressing cells relative to the control. Error bars represent mean ± standard deviation (*n* = 3).

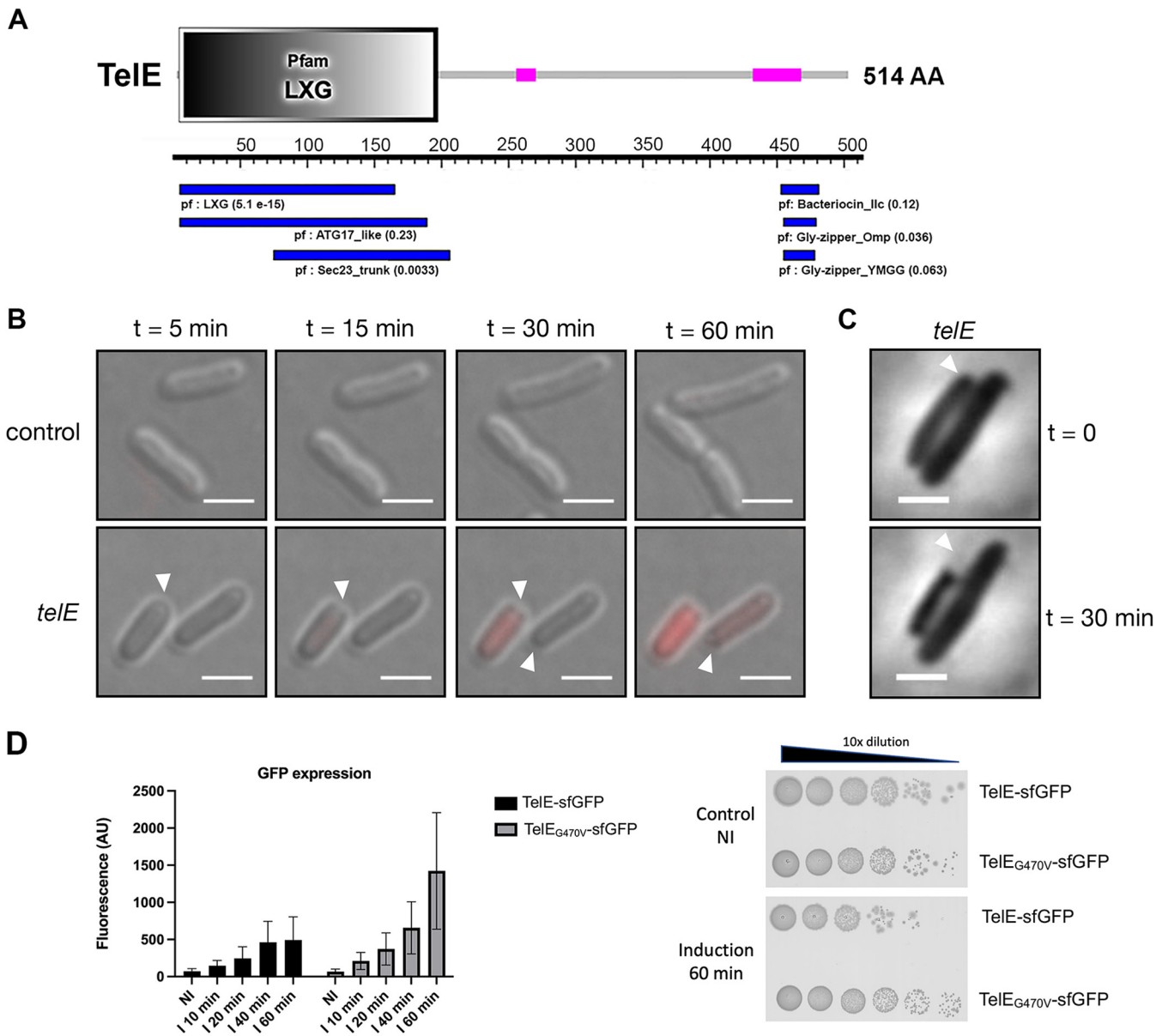

**FIG 4** TelE is a membrane-destabilizing toxin. (A) The Pfam domains found in TelE protein using SMART (top) and motif search from KEGG gene databases (bottom). The pink boxes indicate low-complexity region. The number in parenthesis next to the Pfam domain indicates the E value score. AA, amino acid. (B) Differential interference contrast micrograph of *E. coli* cells harboring an empty vector or a vector encoding *telE* with the expression inducible by anhydrotetracycline (aTc). The physiological changes of the *E. coli* cells following protein induction were documented at the indicated time. Cells with membrane lesions were stained red by propidium iodide. Arrows indicate the region with visible cell shrinkage. Scale bar, 2 μm. (C) Phase contrast micrograph of *telE*-expressing *E. coli* documented at the indicated time. Arrows indicate the region with visible cell shrinkage. Scale bar, 2 μm. (D) Monitoring of TelE expression through translational reporter constructs TelE-sfGFP or TelE$_{G470V}$-sfGFP expressed in *E. coli* under the control of PtetO promoter. GFP fluorescence signal was measured at different time points following anhydrotetracyclin addition (I 10 min to I 60 min) and compared to the initial noninduced (NI) culture. AU, arbitrary units. Cell viability was also monitored in parallel and is shown for the 60-min induction time point.

sfGFP fluorescence level was always higher than TelE-sfGFP. Induction of TelE-sfGFP in *E. coli* for 60 min led to a 2-log decrease in CFU counts compared to the noninduced strain or the strain expressing the mutant version of *telE*, namely, TelE$_{G470V}$-sfGFP (Fig. 4D). This result demonstrates that TelE$_{G470V}$ mutant is produced as well as TelE and that its loss of toxicity is not due to protein degradation.

**The cognate immunity protein TipE antagonizes TelE toxicity.** The activity of LXG family toxins such as TelA, TelB, and TelC can be antagonized by cognate immunity proteins located downstream of the LXG-encoding gene (18). To identify the immunity protein able to counteract TelE toxicity in *E. coli*, each gene located downstream from

*telE* was cloned into a second compatible plasmid (pJN105) in which gene expression was induced by arabinose. We found that only the coexpression of the third gene downstream from *telE* (*gallo_0565*) rescued *E. coli* from TelE intoxication (Fig. 5A; Fig. S5A). This gene encodes a 151-amino-acid-long protein renamed herein as TipE (Tel immunity protein E), which harbors a domain of unknown function DUF5085. An additional DUF5085 domain-containing protein (Gallo_0570), sharing ~30% protein sequence similarity with TipE, was found in five genes downstream of TipE (Fig. 3A). However, coexpression of this protein did not protect *E. coli* from TelE intoxication, suggesting a very specific interaction between TipE and TelE. An AlphaFold prediction model of the TelE-TipE complex is shown in Fig. 5B (TelE in green and TipE in orange). The same model colored with the confidence value is shown in Fig. S6. The predicted interaction of TipE with the region including the glycine-zipper motif of TelE (in yellow) fits well with the inhibitory effect of TipE on TelE.

To demonstrate that TipE interacts with TelE, we constructed a functional, recombinant TelE tagged with a hexahistidyl tagged at the C terminus (TelE-His) and a recombinant TipE tagged with the hemagglutinin tag (HA-TipE) at the N terminus to facilitate the detection of these proteins using an anti-His and anti-HA monoclonal antibody, respectively. Of note, the addition of a hexahistidyl tagged at the C terminus of TelE reduces its toxicity toward *E. coli* compared to the same tag added at the N terminus (Fig. S5C). Coexpression of HA-TipE clearly rescued the growth of *E. coli* in liquid medium (Fig. S5A) as well as TelE-His level as shown in Fig. S5B. Next, a copurification experiment was carried out using nickel-immobilized resin that can retain TelE-His. Incubation of TelE-bound resin with cellular lysate containing HA-TipE captured a significant amount of TipE, which was detected through immunoblot using anti-HA specific antibodies following TelE elution (Fig. 5C). Importantly, when using nontagged TelE as control, HA-TipE was not retained on the free nickel affinity column (Fig. 5C), indicating that TipE indeed interacts with TelE.

Subsequently, we monitored TelE expression in living cells by observing the *E. coli* cells expressing TelE-sfGFP under fluorescence microscopy. Expectedly, TelE expression in *E. coli* cells resulted in the cellular influx of propidium iodide (Fig. 5D). However, instead of being uniformly distributed throughout the cells as the native sfGFP (Fig. 5D), TelE-sfGFP was found in small foci distributed around the cell body (Fig. 5D). In contrast, in *E. coli* cells coexpressing TelE-sfGFP and TipE, TelE lost its distribution as foci in a majority of cells (Fig. 5D). Notably, a small cell population coexpressing TelE-sfGFP and TipE remained stained red by propidium iodide, consistent with our earlier observation that TipE only partially rescued cells from TelE intoxication (Fig. 5A). Taken together, these data showed that TipE is a bona fide immunity protein to TelE, counteracting TelE activity through protein-protein interaction.

## DISCUSSION

In this study, we show that a T7SSb cluster is a putative pathogenicity island conserved in all the clinical isolates of *Streptococcus gallolyticus subsp. gallolyticus* (*SGG*). Interestingly, two different genetic arrangements of the T7SSb machinery can be found in *SGG* clinical isolates together with a different set of putative effectors as shown by comparing the closely UCN34 and TX20005 (Fig. S2). Extensive diversity in T7SS effector repertoires within a given species has been demonstrated in many other T7SSb systems, e.g., *Staphylococcus aureus* (37), *Staphylococcus lugdunensis* (38), *Listeria monocytogenes* (39), and *Streptococcus agalactiae* (40). Similarly, LXG toxin fragments have previously been observed in *S. aureus* and *L. monocytogenes* (41), in *Enterococcus faecalis* (42), and in *S. agalactiae* (43).

The most prevalent T7SS machinery arrangement found in UCN34 resembles that found in *Streptococcus intermedius*, whereas the T7SS of a small minority of *SGG* isolates like TX20005 is identical to the one found in *Staphylococcus aureus* (16, 18) (Fig. S1b). Of note, a second copy of the gene *essC* encoding the motor ATPase is present in TX20005 (Fig. S2). Six T7SSb effectors were identified experimentally in strain UCN34, including the prototypical EsxA/ESAT-6, and three LXG-containing proteins. Among

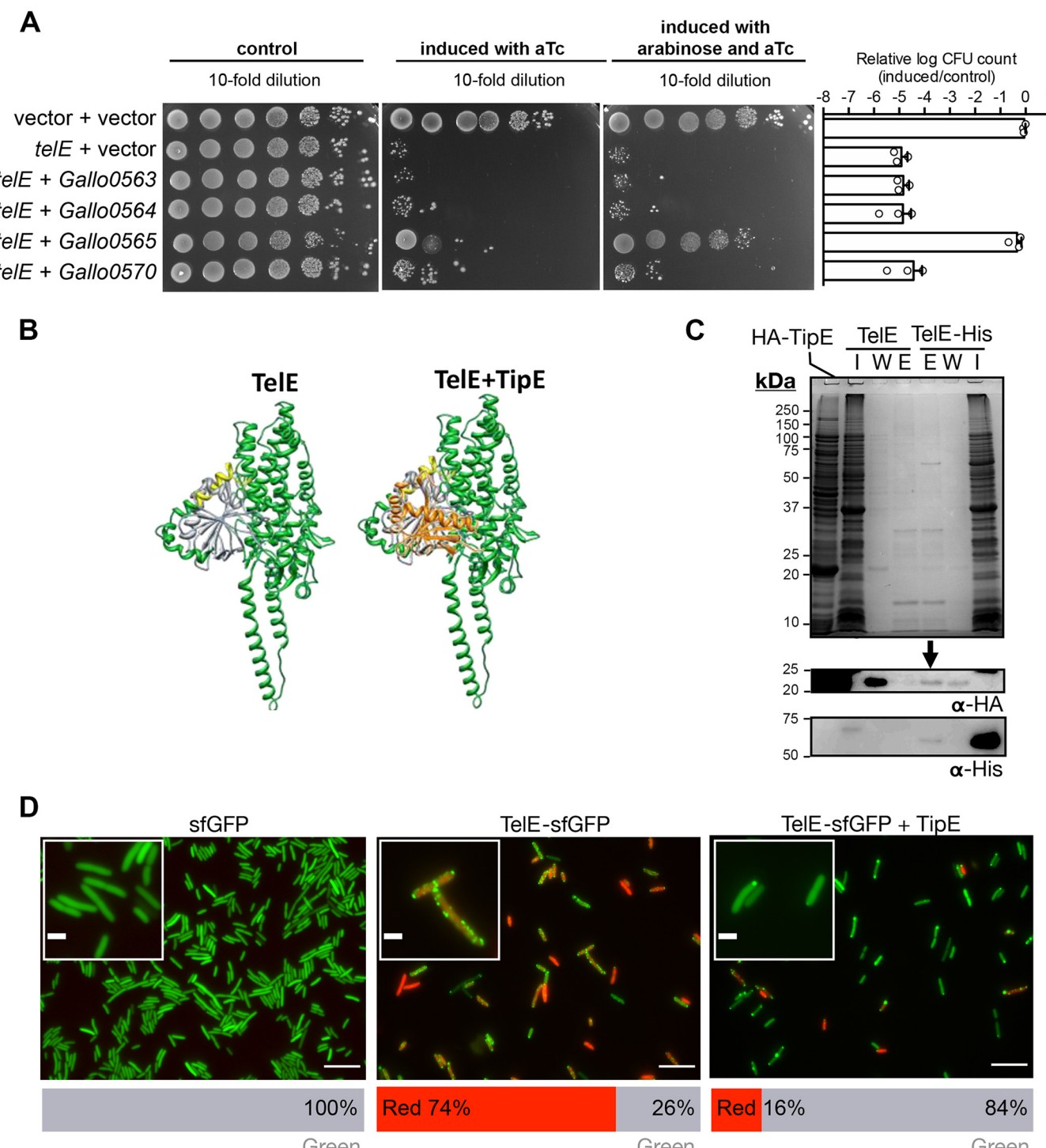

**FIG 5** The cognate immunity protein TipE antagonizes TelE toxicity. (A) Viability of *E. coli* cells harboring two vectors (pTCVerm-P$_{tetO}$ and pJN105-P$_{BAD}$) for single expression of *telE* inducible by anhydrotetracycline (aTc), or coexpression of *telE* and the indicated gene candidates with the expression inducible by arabinose. Bar chart on the right shows the logarithm-transformed CFU count of the cells coexpressing two genes relative to the control. Error bars represent mean ± standard deviation (*n* = 3). (B) AlphaFold modeling of monomeric TelE (in green) without or with TipE (in orange). TipE is predicted to interact with the TelE zipper motif shown in yellow. (C) Immunoblot with anti-His or anti-HA on the proteins eluted from the TelE-bound nickel-immobilized resins incubated with cellular lysate containing HA-TipE. The TelE-free resins subjected to the same treatment were used as the control. I, input (cellular lysate containing HA-TipE); W, wash fraction; E, eluted fraction. (D) Fluorescence micrograph of *E. coli* cells expressing superfolder GFP (sfGFP) or TelE-sfGFP or coexpressing TipE and TelE-sfGFP. Cells with membrane lesions were stained red by propidium iodide. Scale bar, 10 μm; 2 μm for inlets. Bar chart (bottom) shows the quantification of cells fluoresced green or red (*n* ≈ 300).

the three LXG effectors, two encoded outside the T7SSb locus are homologs of TelC, a lipid II phosphatase shown to inhibit peptidoglycan biosynthesis in Gram-positive bacteria (18, 44). The third LXG effector of unknown function is unique to *SGG* and is encoded in the vicinity of the locus encoding T7SSb machinery. Here, we showed that this LXG effector named TelE is toxic when overexpressed in *E. coli*. Similar to other LXG effectors, TelE toxicity in *E. coli* can be neutralized by a cognate immunity protein named TipE.

TipE is an immunity protein of 151 amino acids. In other pore-forming toxins containing glycine zipper motifs such as CdzC/D from *Caulobacter crescentus* and Tse4 from *Pseudomonas aeruginosa*, the cognate immunity proteins are multipass transmembrane proteins residing predominantly in the inner membrane (33, 45). Similarly, an inner membrane protein protects the *E. coli* cells expressing the *S. aureus* membrane depolarizing T7SSb toxin TspA (25). Remarkably, the six-pass transmembrane protein Gallo_0563 encoded by the gene located immediately downstream of *telE* did not protect *E. coli* from TelE intoxication (Fig. 5A and B), and our data showed that only TipE, which contains no transmembrane domain, is able to block TelE activity through direct interaction. Notably, TipE is a DUF5085 family protein predicted related to T7SS in the Pfam database since it is observed in *S. aureus* T7SS (41). Based on our data, we propose herein that DUF5085 family proteins could be a new class of immunity protein to T7SSb toxins. The fact that only the cognate TipE, and not another protein belonging to the same DUF5085 family such as Gallo_0570, confers protection against TelE killing is reminiscent of specific EsaG protection from EsaD killing in *S. aureus* (16).

Interestingly, a diverse pool of TelE variants (TelE to TelE7) exists in the genomes of *SGG* isolates, all harboring the conserved C-terminal glycine zipper motif essential for TelE activity. The glycine zipper motif is a common feature in membrane proteins displaying pore-forming activity. In addition to the aforementioned CdzC/D and Tse4, the glycine zipper motif is also found in the vacuolating toxin VacA from *Helicobacter pylori*, the prion protein (PrP), and the amyloid beta peptide (A$\beta$) (32), many of which display pathological roles. For instance, VacA is involved in gastric ulceration caused by *H. pylori*, whereas PrP and A$\beta$ are associated with prion and Alzheimer's disease, respectively (32). In line with previous observations (32, 33, 46), mutation of the key glycine residue in this motif abolishes TelE activity. Since the glycine residues of the motif are usually positioned in the inner part of membrane pores, we speculate that a change of the small glycine to a bulkier valine residue may affect the oligomerization and/or pore opening of TelE, as demonstrated for other glycine zipper toxins (32).

In addition to TelE, we discovered two additional yet to be characterized T7SSb effectors, namely, Gallo_0559 and Gallo_0560, located immediately upstream of TelE and probably encoded in the same operon. Bioinformatic analysis showed that Gallo_0559 is a DUF5082 family protein whereas Gallo_0560 is a TIGR01497 family protein. Proteins belonging to these two families were previously identified in *S. intermedius* as WxgA and WxgC (recently renamed to LapC1), respectively (18, 28). WxgA and WxgC/LapC1 are chaperone proteins essential for the secretion of LXG family proteins (Tel) in *S. intermedius* (18). These proteins were suggested as WXG100-like proteins (18), but WxgA and another chaperone protein WxgB are 40% longer than the canonical WXG100 proteins. These chaperones have not been detected in the *S. intermedius* bacterial supernatant and were henceforth hypothesized to disassociate from Tel proteins before or during secretion (18). Here, we found that these probable Wxg-like chaperones were detected in the supernatant of *SGG* UCN34 WT but not in the Δ*essC* mutant, indicating that the secretion of these proteins is T7SSb dependent and that they may remain associated with LXG toxins postsecretion. Further investigation on the potential interaction of these two proteins (Gallo_0559 and Gallo_0560) with TelE may provide insight into T7SSb effector secretion and delivery. Similar to the detection of secreted Gallo_0559 and _0560 in this study, a recent preprint indicates that *S. aureus* WXG100/WXG-like EsxBCD proteins interact with T7SS nuclease EsaD and are secreted (47). Klein et al. (28) also showed that one or two LXG-associated proteins located

upstream of the LXG effector are involved in the correct targeting of the LXG effector to the type VII secretion machinery.

The three LXG family toxins identified in this study expand our knowledge of the antibacterial arsenal possessed by *SGG*. It is noteworthy that *SGG* also produces bacteriocins such as gallocin A exhibiting antimicrobial activity against gut commensals such as *E. faecalis* (48), which together with the newly identified Tel toxins, are likely the arsenal essential for *SGG* survival in the highly dynamic and populated human gut environment.

Finally, our findings may have further implications for understanding the pathogenicity of *SGG*. Indeed, it was shown in *SGG* strain TX20005 that T7SSb is a key component contributing to *SGG* gastrointestinal tract colonization and cancer-promoting capability (30). However, the underlying molecular mechanisms are not known. Recently, the same group reported the identification of a 12-gene chromosomal locus named SPAR (for *Sgg* pathogenicity-associated region) reducing the adherence of TX20005 to the host colonic cells and abrogating its ability to stimulate cell proliferation (49). The SPAR mutant phenocopies the T7SSb knockout mutant, connecting the SPAR locus to the T7SSb. Indeed, T7SSb transcription was abolished in the SPAR mutant. Of note, TX20005 belongs to the small minority of *SGG* isolates displaying a different T7SSb gene locus arrangement together with a different repertoire of effectors compared to UCN34, and the impact of these differences on the pathogenic outcome is currently unclear. The contribution of pathogenicity-associated islands in bacterial virulence is well established. For instance, the CagA toxin, VacA toxin, and type IV secretion system encoded in the *Helicobacter pylori* pathogenicity island (50, 51) or the colibactin synthase genes encoded in the *E. coli* polyketide synthase (*pks*) genotoxic island (52–54) are key to the progression of gastric cancer and colorectal cancer, respectively. Of particular interest is VacA, which also contains a glycine zipper motif essential for its function causing ulceration and gastric lesions in the murine model (55). Whether TelE could also alter host cells is a perspective for future studies.

## MATERIALS AND METHODS

**Bacterial culture conditions.** All bacterial strains and plasmids used in this study are listed in Table S1. Cultures of *E. coli* strains were prepared from single colonies in lysogeny broth (LB; BD) and incubated overnight at 37℃, 200 rpm, whereas cultures of *Streptococcus agalactiae* and *Streptococcus gallolyticus* strains were prepared from single colonies in brain heart infusion broth (BHI; Acumedia) and incubated statically overnight at 37℃. When the use of growth medium with a minimal amount of peptides was necessary, overnight cultures of *SGG* were prepared from single colonies in M9YEG broth (1× M9 minimal salts supplemented with 0.5% yeast extract and 1.0% of glucose) (56) and incubated statically at 37℃. When appropriate, antibiotics were supplemented at the following concentrations: kanamycin at 50 $\mu$g/mL for Gram-negative bacteria and 500 $\mu$g/mL for Gram-positive bacteria; gentamicin at 20 $\mu$g/mL for *E. coli*; and erythromycin at 2 $\mu$g/mL for Gram-positive bacteria.

**Genetic techniques.** Standard molecular cloning procedures as previously described (57) were followed to construct plasmids for gene deletion and gene expression. All primers used in this study were synthesized by Integrated DNA Technology and listed in Table S1. The enzymes Q5 DNA polymerase and restriction enzymes were obtained from New England Biolabs, whereas T4 ligase was obtained from Thermo Fisher Scientific. All genetic constructs were delivered into chemically transformed *E. coli* DH5$\alpha$ prepared as described previously (58) and verified by Sanger sequencing performed at 1st BASE. For gene expression plasmids construction, the corresponding open reading frames with the native ribosomal binding sites were cloned downstream of inducible promoters such as P$_{tetO}$ (anhydrotetracycline inducible) and P$_{BAD}$ (arabinose inducible).

To construct plasmids for gene deletion, two ~500-bp DNA fragments corresponding to the upstream and downstream regions of the gene of interest were first PCR amplified. The resulting two DNA fragments were subsequently spliced by overlap extension PCR and ligated into pG1 plasmid at the *Sma*I restriction site. The constructed plasmids were transformed into *E. coli* and later transformed into *S. gallolyticus* as described in the transformation section below. Subsequently, *SGG* in-frame chromosomal deletion mutants were generated by the allelic exchange as previously described (59).

To generate plasmids encoding TelE with a specific residue substitution, *telE* was PCR amplified into two fragments corresponding to the 5′ and 3′ regions by using primers with the desired mutation incorporated. These two fragments were spliced into a full-length *telE* open reading frame by overlap extension PCR and ligated into pTCVerm-P$_{tetO}$ at the *Bam*HI/*Sph*I restriction site.

Plasmid encoding TelE with the C terminus fused with superfolder GFP was constructed by splicing the PCR amplicons corresponding to the *telE* open reading frame without a stop codon and the superfolder GFP open reading frame that has been codon optimized for *SGG* (gene synthesized at GenScript).

**Transformation.** *E. coli* DH5$\alpha$ was chemically transformed as described in reference 58. *S. agalactiae* NEM316 was transformed by electroporation as previously described (60–62). *S. agalactiae* strains carrying the respective plasmids were used for conjugal transfer into *SGG* as described previously (59). *SGG* was also naturally transformed using a newly established method, by supplementing the log-phase culture grown in M9YEG with 10 $\mu$M of the competence inducing peptide (ITGWWGL) outlined in the previous study (63), at least 5 $\mu$g of plasmid DNA, 10 $\mu$M magnesium chloride, 10 $\mu$M magnesium sulfate, and 1 $\mu$M ferrous (II) chloride. The mixture was incubated for an hour at 37°C, followed by 3 h at 30°C, before being plated onto BHI agar supplemented with erythromycin.

**Genomic DNA isolation.** *S. gallolyticus* genomic DNA was extracted using either Wizard Genomic DNA purification kit or Wizard SV Genomic DNA purification kit (Promega). The extraction was performed according to the manufacturer's instructions, except that 20 U to 50 U of mutanolysin (Sigma-Aldrich) was added at the lysozyme treatment step. When necessary, an additional phenol:chloroform clean-up step was performed to remove the excessive impurities in the samples.

**Genomic DNA sequencing and assembly.** High-integrity DNA was sequenced to at least a coverage of 100$\times$ on the Illumina Miseq platform (reagent kit v3, 2 $\times$ 300 bp) at the in-house next-generation sequencing facility in the Singapore Centre for Environmental Life Sciences Engineering. The paired-end sequencing reads were trimmed and assembled *de novo* into contigs on the CLC Genomics Workbench 10.0 (Qiagen). These assembled contigs were subsequently uploaded to the RAST server (64–66) for open reading frame prediction and gene annotation. Raw sequencing reads used in this study are available on NCBI under accession number PRJNA762634.

**Phylogenetic analyses.** A core genome-based alignment was performed on Parsnp version 1.2 integrated in the Harvest suite (67), with *SGG* UCN34 (GenBank accession no. FN597254) serving as the reference genome. The other *S. gallolyticus* complete closed genomes available on NCBI, i.e., *SGG* ATCC 43143 (GenBank accession no. NC_017576), ATCC BAA_2069 (GenBank accession no. NC_015215), and DSM 16831 (GenBank accession no. NZ_CP018822); *SGP* ATCC 43144 (GenBank accession no. NC_015600), NCTC13784 (GenBank accession no. NZ_LS483462), and WUSP067 (GenBank accession no. NZ_CP039457); and two additional draft genomes, *SGM* ACA-DC-206 (unpublished genome) and *SGG* TX20005 (NCBI Assembly ID GCA_000146525), were also included in the analysis for a clear distinction of these closely related strains. The resulting alignment was subsequently visualized on Gingr (integrated in the Harvest suite) and used for phylogenetic inference on FigTree2 based on the approximate-maximum-likelihood algorithm (68).

**Comparative genomics.** A genome-wide comparison among *SGG* UCN34, three representative *SGG* clinical isolates, and three representative *SGP* was performed using BLAST Ring Image Generator v0.95 based on BLASTn search (69). Genomic islands of the *SGG* UCN34 were downloaded from IslandViewer 4 (70). NCBI tBLASTn program was used to identify the protein homologs of the T7SS core components across all the *SGG* and *SGP* clinical isolates. Easyfig v2.2.2 was used to visualize the region of interest and to perform pairwise comparisons on the protein sequences using tBLASTx program (71).

**Comparative proteomics on the *SGG* secretomes.** Single colonies of *SGG* UCN34 WT and the Δ*essC* mutant were inoculated into 20 mL of M9YEG and grown to an optical density at 600 nm (OD$_{600nm}$) of 3.0 corresponding to the early stationary phase of growth. The culture supernatant was collected and filtered through 0.22-$\mu$m filter units. Fifteen milliliters of the filtered culture supernatant was first concentrated using Amicon Ultra 15 mL Centrifugal Filters (cutoff, 10 kDa; Merck), and the proteins were precipitated out of the solution using trichloroacetic acid (TCA) as previously described (72). The samples were further processed and analyzed at the Taplin Mass Spectrometry Facility. Briefly, precipitated proteins were first treated with trypsin (Promega) in 50 mM ammonium bicarbonate solution, were dried in a speed-vac, and were subsequently reconstituted in HPLC solvent A (2.5% acetonitrile and 0.1% formic acid). Peptides were loaded using a Famos autosampler (LC Packings) into a nanoscale reverse-phase HPLC fused silica capillary column (100-$\mu$m inner diameter $\times$ ~30-cm length) packed with 2.6 $\mu$m C$_{18}$ spherical silica beads and eluted with increasing concentrations of solvent B (97.5% acetonitrile, 0.1% formic acid). Eluted peptides were subjected to electrospray ionization and entered into an LTQ Orbitrap Velos Pro ion-trap mass spectrometer (Thermo Fisher Scientific). Peptides were detected, isolated, and fragmented to produce a tandem mass spectrum of specific fragment ions for each peptide. Peptide sequences (and hence protein identity) were determined by comparing the fragmentation pattern acquired by the software program Sequest (Thermo Fisher Scientific) (73) to the protein sequences on protein databases. All databases include a reversed version of all the sequences, and the data were filtered to between a 1% and 2% peptide false discovery rate.

Downstream data analysis was performed in R studio using the differential enrichment analysis of Proteomics Data set 1.10.0 package (74). Proteins that were not detected across all three replicates of either the UCN34 WT or the Δ*essC* mutant were excluded from the analysis. Next, the total peptide count of the remaining proteins ($n = 162$) was scaled and variance stabilized (75) before the missing values (i.e., peptide count $= 0$) were imputed with the assumption that the missing values are missing not at random (MNAR), using the deterministic minimum imputation (MinDet) strategy ($10^{-4}$ quantile) (76). The MinDet-imputed data were subsequently used for differential expression analysis, with a cutoff false discovery rate of 5% and a log$_2$-fold change of 1.

**Bacterial toxicity experiments.** For an intracellular expression of the *SGG* T7SSb effectors, *E. coli* DH5$\alpha$ or GM48 was transformed with pTCVerm-P$_{tetO}$ plasmid series (see Table S1 for the plasmids used in this study). Overnight cultures of these strains were serially diluted in 10-fold increments and 5 $\mu$L of each dilution was spotted onto LB agar supplemented with an appropriate amount of antibiotics and inducer (kanamycin, 50 $\mu$g/mL; anhydrotetracycline, 100 to 500 ng/mL). Agar plates incubated overnight at 37°C were documented on the Gel Doc XR+ Documentation System (Bio-Rad).

For the identification of the TelE cognate immunity protein, the *E. coli* DH5$\alpha$ carrying pTCVerm-P$_{tetO}$::TelE was transformed with compatible plasmids of either pJN105 (vector control) or pJN105 containing each individual open reading frame found downstream of *telE*. The expression of these open reading frames was driven by arabinose-inducible promoter (P$_{BAD}$). Overnight cultures of these strains were first diluted 50 times into sterile LB broth supplemented with antibiotics and incubated at 37°C with agitation at 200 rpm for 1.5 h. These bacterial cultures at the early log-phase of growth were supplemented with arabinose to a final concentration of 0.5%, and incubated further for 3 h, before being subjected to serial dilution and spotted onto LB agar supplemented with an appropriate amount of antibiotics and inducer (kanamycin, 50 $\mu$g/mL; gentamicin, 20 $\mu$g/mL; anhydrotetracycline, 100 ng/mL; and arabinose, 0.5%). Agar plates incubated overnight at 37°C were documented on the Gel Doc XR+ Documentation System (Bio-Rad).

**In silico TelE prediction and homologs identification.** The TelE amino acid sequence from the reference strain *SGG* UCN34 (protein ID CBI13054.1) was uploaded onto the web-based I-TASSER server (77–79) for protein structural prediction using the default settings. The same amino acid sequence was also used to identify TelE homologs among the clinical isolates using the NCBI tBLASTn program. Hit proteins were considered TelE homologs if significant similarities (E value $<10^{-11}$) were identified at the N terminus, C terminus, or full-length protein.

**Multiple sequence alignment and sequence logo generation.** Multiple sequence alignment (MSA) was performed on the online MAFFT server version 7 (https://mafft.cbrc.jp/alignment/server/) with E-INS-i iterative refinement method that assumes all the sequences shared the same conserved motifs (80, 81). Regional sequence logo (probability-weighted Kullback-Leibler logo type) was generated on a web-based Seq2Logo 2.0 server (http://www.cbs.dtu.dk/biotools/Seq2Logo/), with the heuristics clustering method applied (82).

**Fluorescence microscopy imaging.** For time-lapse microscopy imaging, log-phase *E. coli* (OD$_{600nm}$ = ~1.0 Abs) cultured in LB was mixed with propidium iodide (1:1000 diluted from 20 mM stock solution; Thermo Fisher Scientific) and anhydrotetracycline at a final concentration of 500 ng/mL (Sigma-Aldrich) and spotted onto a 1% (wt/vol) low melting agarose pad (bioWORLD). Samples were immediately imaged with the microscope Axio Observer 7 fitted with Plan Apochromat ×40/0.95 dry lens for phase contrast imaging, or Plan Apochromat ×100/1.4 oil lens for differential interference contrast imaging (Carl Zeiss), with temperature maintained at 37°C, and aided with Definite Focus.2 for autofocus. When necessary, samples were excited with filter wheel 555/30 to detect red fluorescence signals.

To image *E. coli* expressing TelE-sfGFP with or without TipE, overnight cultures were diluted 50 times into sterile LB broth, and incubated at 37°C with 200 rpm for 1.5 h to grow to the early log phase. Samples were first induced with 0.5% arabinose for 1.5 h and were subsequently supplemented with anhydrotetracycline to a final concentration of 500 ng/mL and propidium iodide (1:1,000 diluted from 20 mM stock solution, Thermo Fisher Scientific), and incubated further for 2 h at 37°C without agitation. Following incubation, samples were spotted onto a 1% low melting agarose pad (bioWORLD) and imaged with the microscope Axio Observer Z1 fitted with Plan Neofluar ×100/1.3 oil lens, with excitation at 488 nm (GFP) and 535 nm (propidium iodide).

**Flow cytometry.** To analyze TelE-gfp expression, 3 mL of bacterial culture in LB medium was prepared (initial OD around 0.1) from fresh agar plates and grown until reaching an OD$_{600}$ = 0.4 to 0.5. At that time, we removed an aliquot of 500 $\mu$L where no inducer was added (noninduced sample). Anhydrotetracycline at 200 ng/mL was added to the rest of the culture and 500-$\mu$L aliquots were removed at various time points from 30 to 90 min, Hoescht 33342 was directly added to the medium (1/1,000 dilution), and the tubes were left open and incubated 10 min under gentle agitation. After washing and resuspension in PBS, samples were acquired on a MACSQuant YGV Analyzer Apparatus (Milteyni Biotec) and data were analyzed using FlowJo 10 software. GraphPad Prism 9 was used for statistical analysis.

**Protein expression, purification, and copurification.** Protein samples were in general prepared as follows: overnight cultures of *E. coli* BL21(DE3) harboring the respective protein expression pET28b plasmids were diluted 50 times into sterile LB broth supplemented with 50 $\mu$g/mL of kanamycin and grown to an optical density of ~0.7 Abs measured at 600 nm. Subsequently, cultures were added with isopropyl $\beta$-D-1-thiogalactopyranoside (IPTG) to a final concentration of 100 $\mu$M and incubated for an additional 3 h at 37°C, 200 rpm. Following incubation, cells were harvested by centrifugation at 8,000 × *g* for 10 min, and the resulting cell pellets were stored at −80°C (for protein purification). When needed, cells were resuspended in 100 mM HEPES pH 7.5 and lysed by sonication (10s on-off pulse cycles, at 20 or 40% amplitude; Sonics Vibracell VCX750). Unbroken cells were removed by centrifugation at 8,000 × *g* for 10 min at 4°C, and the resulting supernatant was ultracentrifuged (Beckman Coulter XPN100) at 35,000 rpm for 1 h at 4°C to separate the soluble (supernatant) and membrane proteins (pellet).

For protein purification, the membrane pellets were resuspended to a final concentration of 100 mg/mL in buffer A (100 mM HEPES pH 7.5, 500 mM NaCl, and 1% n-dodecyl-$\beta$-D-maltoside [DDM]) and incubated for 1 h at 4°C aided with top-down rotation. The solubilized membrane proteins were subsequently supplemented with imidazole to a final concentration of 50 mM and incubated with nickel-immobilized HisLink resins (Promega) that had been precalibrated in buffer B (100 mM HEPES pH 7.5, 500 mM NaCl, 50 mM imidazole, and 0.1% DDM), for 30 min at 4°C with top-down rotation. Protein-bound HisLink resins were washed with 30× resin volume of ice-cold buffer B and finally rinsed with ice-cold buffer containing 100 mM HEPES pH 7.5, 500 mM NaCl, 500 mM imidazole, and 0.1% DDM to elute the proteins.

For protein pulldown, the protein-bound HisLink resins were incubated with cellular lysate containing HA-TipE prepared in buffer B for 20 min at 4°C aided with top-down rotation. The resins were subsequently washed and the bound proteins were eluted as described above.

**SDS-PAGE and Western blot.** Protein samples were mixed with an equal amount of 2× Laemmli reducing buffer and boiled for 10 min before being subjected to SDS-PAGE analysis on 12% polyacrylamide gels. Proteins were stained with InstantBlue solution (Expedeon) for visualization or were transferred onto PVDF membranes using iBlot Dry Blotting System (Thermo Fisher Scientific) at 20 V for 7 min. Following protein transfer, the PVDF membranes were immediately blocked with 1× casein blocking buffer (Sigma-Aldrich) and were probed with either horseradish peroxidase (HRP)-conjugated primary antibodies against His tag (Qiagen) or with rabbit primary antibodies against HA-tag (Sigma-Aldrich) followed by anti-rabbit HRP-conjugated secondary antibodies. Membranes were thoroughly rinsed with PBS containing 0.1% Tween 20 between the incubations. Chemiluminescent signals were detected using Amersham ImageQuant 800 System (GE Healthcare) after a brief incubation of the PVDF membranes with Immobilon Forte Western Chemiluminescent HRP Substrates (EMD Millipore). For reblotting with a different antibody after image acquisition, the membranes were incubated with Restore PLUS Western Blot Stripping Buffer (Thermo Fisher Scientific) for 30 min at room temperature, rinsed thoroughly before being blocked with casein blocking buffer, and probed with antibodies as described above.

## SUPPLEMENTAL MATERIAL

Supplemental material is available online only.
**SUPPLEMENTAL FILE 1**, PDF file, 4.5 MB.
**SUPPLEMENTAL FILE 2**, DOCX file, 0.02 MB.
**SUPPLEMENTAL FILE 3**, DOCX file, 0.03 MB.
**SUPPLEMENTAL FILE 4**, XLSX file, 0.05 MB.

## ACKNOWLEDGMENTS

We are very grateful to Alexandra Doloy and Nicolas Dmytruk for providing us with the collection of *S. gallolyticus* clinical isolates. We thank Ross Tomaino from Taplin Mass Spectrometry Facility for the mass-spectrometry analysis, Daniela Megrian-Nunez for her help with AlphaFold, and Bruno Périchon for his help in the comparative analysis of the T7SSb genetic locus shown in Fig. S1b. We thank Tarek Msadek for critical reading of the manuscript.

This work was supported by the National Research Foundation and Ministry of Education Singapore under its Research Centre of Excellence Program (SCELSE). S. Dramsi acknowledges the support of the Institut National contre le Cancer (INCA; grant PLBIO16-025) and from the French Government's Investissement d'Avenir program, Laboratoire d'Excellence Integrative Biology of Emerging Infectious Diseases (grant no. ANR-10-LABX-62-IBEID). A. Filloux acknowledges the support of Singapore Centre for Environmental Life Sciences Engineering (SCELSE; grant no. 04MNS001779A660OOE01).

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
