## [Reviewer comments · Microbiology Spectrum]

Microbiology Spectrum

Characterization of Tele, a T7SS LXG effector exhibiting a conserved C-terminal glycine zipper motif required for toxicity

Wooi Keong TEH, Yichen Ding, Francesca Gubellini, Alain Filloux, Claire Poyart, Michael Givskov, and Shaynoor Dramsi

Corresponding Author(s): Shaynoor Dramsi, Institut Pasteur

Review Timeline:

Submission Date:	April 7, 2023
Editorial Decision:	May 10, 2023
Revision Received:	June 14, 2023
Accepted:	June 22, 2023

Editor: Christopher LaRock

Reviewer(s): The reviewers have opted to remain anonymous.

Transaction Report:

DOI: <https://doi.org/10.1128/spectrum.01481-23>

May 10, 2023

Dr. Shaynoor Dramsi
Institut Pasteur
28 rue du Dr Roux
Paris
France

Re: Spectrum01481-23 (Characterization of TelE, a T7SS LXG effector exhibiting a conserved C-terminal glycine zipper motif required for toxicity)

Dear Dr. Shaynoor Dramsi:

On the basis of recommendations from expert reviewers in the field, I have determined that your manuscript requires some edits before acceptance. I believe these concerns can be addressed as a "text-only" revision. The reviewers found the study to overall be well-conducted and of interest to the field, but noted a few points for clarification or overreaching statements (e.g. pore-formation, when the experimental evidence is limited to showing membrane disruption and not specifically and directly insertion into a membrane or formation of a discrete pore). While you are revising the manuscript, please specifically address the attached comments from the Reviewers. Additionally, please review the ASM Open Data Policy page for requirements on sequence and other data availability.

Thank you for submitting your manuscript to Microbiology Spectrum. As you will see your paper is very close to acceptance. Please modify the manuscript along the lines I have recommended. As these revisions are quite minor, I expect that you should be able to turn in the revised paper in less than 30 days, if not sooner. If your manuscript was reviewed, you will find the reviewers' comments below.

When submitting the revised version of your paper, please provide (1) point-by-point responses to the issues raised by the reviewers as file type "Response to Reviewers," not in your cover letter, and (2) a PDF file that indicates the changes from the original submission (by highlighting or underlining the changes) as file type "Marked Up Manuscript - For Review Only". Please use this link to submit your revised manuscript. Detailed instructions on submitting your revised paper are below.

Link Not Available

Sincerely,

Christopher LaRock

Reviewer comments:

Preparing Revision Guidelines

- Point-by-point responses to the issues raised by the reviewers in a file named "Response to Reviewers," NOT IN YOUR COVER LETTER.

- Upload a compare copy of the manuscript (without figures) as a "Marked-Up Manuscript" file.
- Each figure must be uploaded as a separate file, and any multipanel figures must be assembled into one file.
- Manuscript: A .DOC version of the revised manuscript
- Figures: Editable, high-resolution, individual figure files are required at revision, TIFF or EPS files are preferred

Please return the manuscript within 60 days; if you cannot complete the modification within this time period, please contact me. If you do not wish to modify the manuscript and prefer to submit it to another journal, please notify me of your decision immediately so that the manuscript may be formally withdrawn from consideration by Microbiology Spectrum.

In this manuscript, Teh et al performed comparative genomics between subspecies of *S. gallolyticus* and found that subspecies *gallolyticus* (Sgg) uniquely encodes T7SS genes, often in a genomic arrangement similar to that found in *S. intermedius*. They further identified six Sgg T7SS effectors including a novel LXG-toxin, TelE, which promotes membrane integrity loss and can be mitigated by immunity factor TipE. Overall, this study provides an interesting and informative elaboration on the Sgg T7SS as well as identification and characterization of a novel T7SSb LXG toxin. Concerns exist that direct experimental evidence is not provided for claims of TelE pore-forming activity, that the nomenclature chosen for Sgg T7SS proteins may cause confusion in the literature, and that relevant T7SSb literature is not sufficiently discussed. Additional concerns are listed below.

Major:

1. Claims of TelE pore-formation are not experimentally substantiated as pore-forming activity/membrane insertion of TelE is not directly shown via planar lipid bilayers or TEM pore visualization of the protein/in liposomes, etc. The description of TelE activity should be modified in the text to indicate that TelE promotes loss of membrane integrity, as indicated by propidium iodide staining. It is very possible this is due to pore formation; however, the evidence provided here is indirect and pore formation is not the only cause of membrane integrity loss.
2. The authors should reconsider the naming of Sgg LXG proteins Gallo_1068 and 1574 as TelC1 and TelC2. This nomenclature may promote confusion of Sgg LXG proteins with *S. intermedius* TelC (which would technically be TelC1 as it was identified first by Whitney et al, *eLife*, 2017). Further, as these Sgg LXG proteins exhibit only 40% similarity to the *S. intermedius* TelC and their toxicity/function as a lipid II phosphatase have not yet been experimentally confirmed, naming them “TelC” in Sgg is premature.

Further, Klein et al (*Mbio*, 2022) recently proposed to rename LXG toxin chaperones proteins as “LXG-associated proteins (Lap)”, replacing the WxgA/WxgB/WxgC nomenclature. *S. intermedius* TelC chaperones are now referred to as LapC1 and LapC2. Naming Sgg LXG toxins TelC1 and TelC2 would complicate this Lap nomenclature and make distinguishing these *S. intermedius* and Sgg chaperones difficult in literature. A nomenclature highlighting specificity to Sgg would be ideal to prevent this confusion.

As a follow-up to the above, how homologous are Gallo_1068 and 1574? Do they structurally resemble each other or *S. intermedius* TelC? Are UCN34 “Laps” homologous to *S. intermedius* LapC1 and LapC2?

3. More discussion of relevant T7SSb literature is needed. Another group recently published on Sgg T7SS effectors encoded within a pathogenicity locus in strain TX20005 (Taylor/Xu et al; *bioRxiv* preprint, April 2022; now published in *Scientific Reports*; April 2023), but this study is not mentioned in the present manuscript. The authors should discuss this work as it is only the second published study on the Sgg T7SS to date. Are the T7SS proteins identified in TX20005 (SparG - SparL) homologous to any T7SS proteins in UCN34?

Additional acknowledgement of previous T7SSb findings would also be helpful throughout the manuscript. The below concepts are related to this work and should be mentioned/discussed.

- Lines 131-133 indicate that putative T7SSb effectors differ across Sgg strains but, aside from the diagram in Fig S2, this is not elaborated on. Extensive diversity in effector repertoires within a given species has been shown in other T7SSb systems: *Listeria monocytogenes* (Bowran/Palmer, *Microbiology*, 2020), *Staph aureus* (Warne et al, *BMC Genomics*, 2016), *Staph lugdunensis* (Lebeurre et al, *Frontiers in Microbiology*, 2019), and *Strep agalactiae* (Spencer, *Plos Path* 2021). Can the authors provide more detail on diversity of Sgg T7SS effector repertoires/subtypes? It would be helpful to compare this to the previously observed intra-species T7SSb diversity.
- T7SS LXG toxin fragments have previously been observed in *S. aureus* (Bowman/Palmer, *Annual Review of Microbiology*, 2021), *E. faecalis* (Chatterjee et al, *PloS Biology*, 2020), *S. agalactiae* (Spencer et al, *bioRxiv*, 2023), and *L. monocytogenes* (Bowran/Palmer, *Microbiology*, 2021).
- Similar to detection of secreted Gallo_0559 and _0560 in this manuscript, a recent preprint indicates that *S. aureus* WXG100/WXG-like EsxBCD proteins interact with T7SS nuclease EsaD and are secreted (Yang/Palmer et al, *bioRxiv*, 2023).

- DUF5085 proteins have been observed in *S. aureus* T7SS (Bowman/Palmer, *Annual Review of Microbiology*, 2021)
- The fact that only the cognate TipE confers protection against TelE killing is reminiscent of specific EsaG protection from EsaD killing in *S. aureus*. (Cao/Palmer et al, *Nature Microbiology*, 2016).
- Citation should be included on line 90.

Minor comments:

- Sequences obtained for clinical isolates should be uploaded to a public repository and accession numbers provided in the manuscript. Information on the 38 Sgg isolates (e.g., body site isolated from) would also be helpful.
- Statistical analyses were not performed in this manuscript. This may not be needed for *E. coli* intoxication assays but would be useful for microscopy data in Figures 4 and 5.
- Introduction: The transition from *S. gallolyticus* phylogeny to T7SS in the opening paragraphs of the introduction would be clearer if comparative genomics were introduced sooner. Can the authors clarify why completion of the UCN34 genome provided “clear insights for its adaptation to the rumen” in lines 62-63? Lines 70-71: SGP is less pathogenic but is common in neonatal meningitis, which seems contradictory.
- Can the authors comment on the effector repertoires encoded downstream of T7SS machinery genes (Fig S2)? Are these the only two Sgg T7SS arrangements of effectors found across the 38 isolates? Are different effector repertoires associated with the different machinery arrangements and/or does effector repertoire/machinery arrangement correlate with strain isolation site/host?
- More information on the conservation/location/genetic arrangement of the *telE* variants would be helpful. Is *telE* always associated with the T7SS locus and separated from *tipE* by the same two genes? Is *telE* always preceded by the same two Lap genes? Of the strains encoding more than one TelE variant (in Fig S3), are those genes encoded proximally to each other/the T7SS locus, or are they orphaned genes encoded elsewhere in the genome?
- Of the TelE point mutant-expressing strains that did not confer toxicity in Fig 3E, were those plasmids sequenced following toxin induction to confirm that an additional suppressor mutation is not responsible for the observed loss of toxicity?
- Lines 88-89; 163-164: I am not aware of a role for some of these toxins in antibacterial activity (TspA, EsxX). LXG toxins have also been implicated in virulence.
- The mention of the “main” ATPase in line 137 sounds as if UCN34 encodes a second *essC*. Is this the case?
- Line 139 should indicate differential abundance in supernatant (rather than expression).
- Line 178: What was the identity cut-off used to identify TelE variants? As TelE4 exhibits very low homology to other TelE proteins (Fig S4; solely in the glycine motif) it is a less convincing TelE homolog.
- Is TelE1 the TelE that is used for experiments in Fig 3E-Fig5?
- The protein gel in Fig 5C should be repeated as the inversion of well order between TelE and TelE-His is not ideal, it appears that two lanes may have run together in the anti-HA blot, and in the anti-His Western blot shows very minimal eluted TelE-His (especially compared to the input) as well as minimal captured TipE. These data would be strengthened if the reciprocal co-IP experiment was also performed, using His-tagged TipE as bait.
- Is *tipE* encoded by any non-pathogenic *S. gallolyticus* strains (to prevent from killing by related subspecies?)

- Line 298. As several T7SS toxins have cytosolic immunity proteins, I would not categorize TipE as an “atypical” or “non-canonical” T7SS immunity protein solely because it does not encode transmembrane domains. It seems more unusual based on previous literature that the immunity factor gene would be encoded so far downstream of the toxin gene. Can the authors comment in the Discussion on other examples of this kind of genetic toxin-immunity factor arrangement in the literature?

Minor comments on tables, figures, and legends:

- Some information may be remnant from a prior version of this manuscript and is no longer relevant to the current manuscript (Lines 380-382, antibiotics listed; Table S1 and lines 417-420, *S. agalactiae*/*E. faecalis* strains listed that are not used in the current manuscript).
- Fig S1B is missing panel A-C designations as mentioned in the legend. What identity cut-off was used to designate a “homolog” for T7SS machinery proteins in Fig S1b-panel B? It is not clear whether the “highly similar” indication in blue refers to the T7SS locus in panel C. Please indicate more clearly in the figure that *esxA* is not encoded directly adjacent to *essA* in the B196 genome.
- Figure S2: it would be helpful to label the UCN34 T7SS genes that encode the identified secreted proteins. Light pink and bright blue arrows in Figure S2 are not identified by the key.
- Fig2B and Fig S2: Gallo_0560 should be annotated as a TIGR4197 family protein as mentioned in line 148.
- Figure 4A: What do the pink boxes indicate?
- The phylogenetic tree in Fig S3 appears to be redundant with that shown in Fig S1. The Fig S3 legend should refer to C-terminal TelE sequences in Figure 3C instead of Fig 3B.
- Figure S3: Proteins are annotated as TelD. In Suppl Table 1, some genes are referred to as *tipD*.
- Figure S5A does not currently show *E. coli* toxicity due to TelE-sfGFP expression as stated in lines 221-222. Think this should refer to the right part of Fig 5C instead? I do not see the current Fig S5A (OD600) referenced in the text at this time.
- Is line 274 meant to reference Fig 5A? In Fig 5A, it appears that TipE fully rescued TelE toxicity (in contrast to the claim made in line 274 regarding partial TipE rescue).
- Line 508: What is/which experiments utilized *E. coli* GM48? If used, it should be added to Table S1.
- Line 957: In Figure 5 legend, the AlphaFold model should be labeled as panel (B).

Summary:

Manuscript by Wooi Keong The et al., titled “Characterization of TelE, a T7SS LXG effector exhibiting a conserved C-terminal glycine zipper motif required for toxicity” investigates Type 7b Secretion System (T7SSb) loci encoded by the *Streptococcus gallolyticus* subsp. *gallolyticus* (SGG). Authors characterize the genetic organization of these loci and find SGG strain UCN34 to secrete 6 substrates via T7SSb. One of these secreted proteins, hereby named TelE, is found to be toxic when overexpressed in *Escherichia coli*. TelE is reminiscent of glycine zipper pore forming proteins and contains conserved glycine rich motif required for TelE toxic activity in *E. coli*. Toxicity of TelE was found to be partially alleviated by TipE that serves as an antitoxin.

This work is of interest to people studying *Streptococcus gallolyticus* or T7SS as well as to the broader audience studying bacterial physiology, pathogenesis and interbacterial antagonism.

Overall, this is a well written manuscript that presents compelling data characterizing a novel type of T7SSb-secreted polymorphic pore-forming toxin. The experimental methods and approaches are appropriate and support authors' conclusions.

Comments:

Page 6, line 116: should “Type VIIb Secretion System” be abbreviated as “T7SSb” or T7bSS?

Page 10, line 222: I believe it should be “Fig. S5C”

Page 11, line 229: Protein degradation of the TelE portion of the fusion protein could still occur?

Page 37, line 957: In the Figure 5 legend, “(B)” is mislabeled as “(C)”

Fig 4B: Perhaps it would be advantageous to quantify the microscopy data presented in Fig. 4B.

Fig 5C: Why isn't HA-tag detected in the Input (I) lanes?

Reviewer 1

*In this manuscript, Teh et al performed comparative genomics between subspecies of *S. gallolyticus* and found that subspecies *gallolyticus* (*Sgg*) uniquely encodes T7SS genes, often in a genomic arrangement similar to that found in *S. intermedius*. They further identified six *Sgg* T7SS effectors including a novel LXG-toxin, *TelE*, which promotes membrane integrity loss and can be mitigated by immunity factor *TipE*. Overall, this study provides an interesting and informative elaboration on the *Sgg* T7SS as well as identification and characterization of a novel T7SSb LXG toxin. Concerns exist that direct experimental evidence is not provided for claims of *TelE* pore-forming activity, that the nomenclature chosen for *Sgg* T7SS proteins may cause confusion in the literature, and that relevant T7SSb literature is not sufficiently discussed. Additional concerns are listed below.*

Major:

*1. Claims of *TelE* pore-formation are not experimentally substantiated as pore-forming activity/membrane insertion of *TelE* is not directly shown via planar lipid bilayers or TEM pore visualization of the protein/in liposomes, etc. The description of *TelE* activity should be modified in the text to indicate that *TelE* promotes loss of membrane integrity, as indicated by propidium iodide staining. It is very possible this is due to pore-formation; however, the evidence provided here is indirect and pore formation is not the only cause of membrane integrity loss.*

We agree and we have revised the manuscript to take into account this point.

*2. The authors should reconsider the naming of *Sgg* LXG proteins Gallo_1068 and 1574 as *TelC1* and *TelC2*.*

*This nomenclature may promote confusion of *Sgg* LXG proteins with *S. intermedius* *TelC* (which would technically be *TelC1* as it was identified first by Whitney et al, eLife, 2017). Further, as these *Sgg* LXG proteins exhibit only 40% similarity to the *S. intermedius* *TelC* and their toxicity/function as a lipid II phosphatase have not yet been experimentally confirmed, naming them “*TelC*” in *Sgg* is premature. Further, Klein et al (Mbio, 2022) recently proposed to rename LXG toxin chaperones proteins as “LXG-chaperones are now referred to as *LapC1* and *LapC2*. Naming *Sgg* LXG toxins *TelC1* and *TelC2* would complicate this *Lap* nomenclature and make distinguishing these *S. intermedius* and *Sgg* chaperones difficult in literature. A nomenclature highlighting specificity to *Sgg* would be ideal to prevent this confusion.*

We agree and have removed *TelC1* and *TelC2* from Fig. 2. We have left the original nomenclature Gallo_1068 and Gallo_1574.

*As a follow-up to the above, how homologous are Gallo_1068 and 1574? Do they structurally resemble each other or *S. intermedius* *TelC*?*

Here are the homologies in detail **for review only**.

- Gallo_1068 vs *TelC* : Coverage of 98%, similarity of 43%, with e-value of 6e-139.
- Gallo_1574 vs *TelC* : Coverage of 99%, similarity 64%, with e-value of 0.
- Gallo_1068 vs Gallo_1574 : Coverage of 99%, similarity of 43%, with e-value of 9e-143.

Below we provide the predictions using AlphaFold coloured based on the pLDDT score (pLDDT =predicted local distance difference test), a reliability per residue measure (blue for high confidence, red for low confidence).

pLDDT-coloured AlphaFold2 models of proteins SGG_1574 and SGG_1068.

Are UCN34 “Laps” homologous to S. intermedius LapC1 and LapC2?

LapC1 and LapC2 do share a certain degree of similarities with Gallo_1576 (45%) and Gallo_1575 (53%), respectively. However, with the renaming of TelC2 back to Gallo_1574, we will not call these proteins as Lap homologs.

3. More discussion of relevant T7SSb literature is needed. Another group recently published on Sgg T7SS effectors encoded within a pathogenicity locus in strain TX20005 (Taylor/Xu et al; bioRxiv preprint, April 2022; now published in Scientific Reports; April 2023), but this study is not mentioned in the present manuscript. The authors should discuss this work as it is only the second published study on the Sgg T7SS to date. Are the T7SS proteins identified in TX20005 (SparG - SparL) homologous to any T7SS proteins in UCN34?

We agree and have now added a few lines about this paper in the discussion (lines 382-388). SparG-SparL are 99% similar at nucleotide level to the region annotated as Gallo_1066 to Gallo_1072.

Additional acknowledgement of previous T7SSb findings would also be helpful throughout the manuscript.

The below concepts are related to this work and should be mentioned/discussed.

- Lines 131-133 indicate that putative T7SSb effectors differ across Sgg strains but, aside from the diagram in Fig S2, this is not elaborated on. Extensive diversity in effector repertoires within a given species has been shown in other T7SSb systems: Listeria monocytogenes (Bowran/Palmer, Microbiology, 2020), Staph aureus (Warne et al, BMC Genomics, 2016), Staph lugdunensis (Lebeurre et al, Frontiers in Microbiology, 2019), and Strep agalactiae (Spencer, Plos Path 2021).*

We have now added references to previous T7SSb findings in the discussion (lines 301-303).

Can the authors provide more detail on diversity of Sgg T7SS effector repertoires/subtypes? It would be helpful to compare this to the previously observed intra-species T7SSb diversity.

The diversity of the T7SSb machinery and repertoire will be detailed in a manuscript currently under preparation describing the genomic comparison of these 40 SGG clinical isolates.

- T7SS LXG toxin fragments have previously been observed in S. aureus (Bowman/Palmer, Annual Review of Microbiology, 2021), E. faecalis (Chatterjee et al, PloS Biology, 2020), S. agalactiae (Spencer et al, bioRxiv, 2023), and L. monocytogenes (Bowran/Palmer, Microbiology, 2021).*

Agreed. We have added these bibliographical references in the discussion (lines 306-309).

- *Similar to detection of secreted Gallo_0559 and _0560 in this manuscript, a recent preprint indicates that S. aureus WXG100/WXG-like EsxBCD proteins interact with T7SS nuclease EsaD and are secreted (Yang/Palmer et al, bioRxiv, 2023).*
- *DUF5085 proteins have been observed in S. aureus T7SS (Bowman/Palmer, Annual Review of Microbiology, 2021)*
- *The fact that only the cognate TipE confers protection against TelE killing is reminiscent of specific EsaG protection from EsaD killing in S. aureus. (Cao/Palmer et al, Nature Microbiology, 2016).*
- *Citation should be included on line 90.*

We agree. This sentence and the citation were added in the discussion (line 342-345).

Minor comments:

- *Sequences obtained for clinical isolates should be uploaded to a public repository and accession numbers provided in the manuscript. Information on the 38 Sgg isolates (e.g., body site isolated from) would also be helpful.*

Sequences are uploaded to NCBI under the accession number PRJNA762634. This information is added in the method section.

- *Statistical analyses were not performed in this manuscript. This may not be needed for E. coli intoxication assays but would be useful for microscopy data in Figures 4 and 5.*

The result of microscopy data quantified in Fig. 5 is so clearcut that we do not think that statistical analyses is necessary.

- *Introduction: The transition from S. gallolyticus phylogeny to T7SS in the opening paragraphs of the introduction would be clearer if comparative genomics were introduced sooner. Can the authors clarify why completion of the UCN34 genome provided “clear insights for its adaptation to the rumen” in lines 62-63?*

SGG genome was found to contain a very high number of enzymes involved in the degradation of complex carbohydrates found in plants but also can detoxify the plants tannins in the rumen of herbivores. We have added a sentence in the introduction.

Lines 70-71: SGP is less pathogenic but is common in neonatal meningitis, which seems contradictory.

You are right. We have changed this in the revised version (line 77).

- *Can the authors comment on the effector repertoires encoded downstream of T7SS machinery genes (Fig S2)? Are these the only two Sgg T7SS arrangements of effectors found across the 38 isolates? Are different effector repertoires associated with the different machinery arrangements and/or does effector repertoire/machinery arrangement correlate with strain isolation site/host?*

All the SGG clinical isolates were isolated from the blood of patients. The T7SS locus of each clinical isolate will be discussed in detail in a future paper on genomic comparisons of these isolates.

- *More information on the conservation/location/genetic arrangement of the telE variants would be helpful. Is telE always associated with the T7SS locus and separated from tipE by the same two genes? Is telE always preceded by the same two Lap genes? Of the strains encoding more than one TelE variant (in Fig S3), are those genes encoded proximally to each other/the T7SS locus, or are they orphaned genes encoded elsewhere in the genome?*

The genomes of the clinical isolates were only sequenced and assembled as draft genomes with multiple contigs. Whereas most of the genomes carry the full region of Gallo0560 (genes preceding Gallo0562 TelE) to Gallo0565 (TipE), this region was not assembled into one single contig in some

genomes. Therefore, we are unable to draw conclusion on the conservation of the region across all the isolates we sequenced. In the same vein, we are unable to comment on the genomes carrying multiple *TeE* homologs.

- *Of the *TeE* point mutant-expressing strains that did not confer toxicity in Fig 3E, were those plasmids sequenced following toxin induction to confirm that an additional suppressor mutation is not responsible for the observed loss of toxicity?*

No, the plasmids were not sequenced following toxin induction. However, the experiment shown in Fig3E was repeated on separate days, using cultures prepared from single colonies. Plasmids isolated from the culture was sequenced to verify that only the desired point mutation was observed. It is unlikely that the observed loss of toxicity is due to suppressor mutation.

- *Lines 88-89; 163-164: I am not aware of a role for some of these toxins in antibacterial activity (*TspA*, *EsxX*). LXG toxins have also been implicated in virulence.*

Membrane depolarizing and intraspecies competition activity of *TspA* had been demonstrated. *EsxX* is indeed implicated in virulence. The sentence is corrected for clarity.

- *The mention of the “main” ATPase in line 137 sounds as if UCN34 encodes a second *essC*. Is this the case?*

No there is only one copy of *essC* in SGG UCN34.

- *Line 139 should indicate differential abundance in supernatant (rather than expression).*

Ok we have changed this.

- *Line 178: What was the identity cut-off used to identify *TeE* variants? As *TeE4* exhibits very low homology to other *TeE* proteins (Fig S4; solely in the glycine motif) it is a less convincing *TeE* homolog.*

All *TeE* homologs were identified using NCBI tblastn. Except *TeE4*, all *TeE* homologs share at least 95% similarities. *TeE4* was identified based on similarity (55%) at the C-terminal with significant hit (e-value of 1.64e-11). Further analysis identified that *TeE4* contains LXG domain and glycine zipper motif. Since glycine zipper motif was identified to be essential of *TeE*'s function, we therefore included *TeE4* as *TeE* homolog.

- *Is *TeE1* the *TeE* that is used for experiments in Fig 3E-Fig5?*

Yes

- *The protein gel in Fig 5C should be repeated as the inversion of well order between *TeE* and *TeE-His* is not ideal, it appears that two lanes may have run together in the anti-HA blot, and in the anti-His Western blot shows very minimal eluted *TeE-His* (especially compared to the input) as well as minimal captured *TipE*. These data would be strengthened if the reciprocal co-IP experiment was also performed, using Histagged *TipE* as bait.*

We are sorry but the first author of this paper is no longer in the laboratory, and we are not able to repeat this experiment.

- *Is *tipE* encoded by any non-pathogenic *S. gallolyticus* strains (to prevent from killing by related subspecies?)*

TipE is not encoded by the *S. gallolyticus* subsp. *pasteurianus*.

- *Line 298. As several T7SS toxins have cytosolic immunity proteins, I would not categorize *TipE* as an “atypical” or “non-canonical” T7SS immunity protein solely because it does not encode transmembrane domains. It seems more unusual based on previous literature that the immunity factor gene would be encoded so far downstream of the toxin gene. Can the authors comment in the*

Discussion on other examples of this kind of genetic toxin-immunity factor arrangement in the literature?

We have deleted the epithet “atypical” or “non-canonical” to qualify TipE throughout the manuscript. We were also very surprised to find the immunity gene so far from the LXG-toxin and we are not aware of other examples of this kind of genetic toxin-immunity factor arrangement in the literature. Plus the literature in the domain of toxin-antitoxin is plethoric which does not facilitate this bioinformatic query.

Minor comments on tables, figures, and legends:

- *Some information may be remnant from a prior version of this manuscript and is no longer relevant to the current manuscript (Lines 380-382, antibiotics listed; Table S1 and lines 417-420, S. agalactiae/E. faecalis strains listed that are not used in the current manuscript).*

OK thank you for your vigilance.

- *Fig S1B is missing panel A-C designations as mentioned in the legend. What identity cut-off was used to designate a “homolog” for T7SS machinery proteins in Fig S1b-panel B? It is not clear whether the “highly similar” indication in blue refers to the T7SS locus in panel C. Please indicate more clearly in the figure that esxA is not encoded directly adjacent to essA in the B196 genome.*

Thank you. We have added A-C labels in Fig. S1B and have marked with two lines that esxA is not adjacent to essA.

- *Figure S2: it would be helpful to label the UCN34 T7SS genes that encode the identified secreted proteins. Light pink and bright blue arrows in Figure S2 are not identified by the key.*

We have omitted non-essential information in the legend.

- *Fig2B and Fig S2: Gallo_0560 should be annotated as a TIGR4197 family protein as mentioned in line 148.*

We have added this information in Fig. 2B

- *Figure 4A: What do the pink boxes indicate?*

Pink boxes indicate low complexity region. We have added this information in the legend.

- *The phylogenetic tree in Fig S3 appears to be redundant with that shown in Fig S1. The Fig S3 legend should refer to C-terminal TelE sequences in Figure 3C instead of Fig 3B.*

We prefer to keep this as it stands. We have corrected the typo Fig. S3 legend and in the Fig. S3 replace TelD by TelE.

- *Figure S3: Proteins are annotated as TelD. In Suppl Table 1, some genes are referred to as tipD.*

Thank you. It has been corrected.

- *Figure S5A does not currently show E. coli toxicity due to TelE-sfGFP expression as stated in lines 221-222.*

Correct we meant Fig. 4D the right part of the figure.

I do not see the current Fig S5A (OD600) referenced in the text at this time.

Correct this has been added in the revised version.

- *Is line 274 meant to reference Fig 5A? In Fig 5A, it appears that TipE fully rescued TelE toxicity (in contrast to the claim made in line 274 regarding partial TipE rescue).*

In Fig. 5A there is still a difference between the TelE-TipE (5th line) as compared to the control (1st line). Thus, it is a partial rescue.

• *Line 508: What is/which experiments utilized E. coli GM48? If used, it should be added to Table S1.*
It was used as host for expression of pTCV-TelEgfp or TelEG470V-gfp. We have add it in Table S1.

• *Line 957: In Figure 5 legend, the AlphaFold model should be labeled as panel (B).*
Done. Thank you for your vigilance.

Reviewer 2

Summary:

Manuscript by Wooi Keong The et al., titled "Characterization of TelE, a T7SS LXG effector exhibiting a conserved C-terminal glycine zipper motif required for toxicity" investigates Type 7b Secretion System (T7SSb) loci encoded by the Streptococcus gallolyticus subsp. gallolyticus (SGG). Authors characterize the genetic organization of these loci and find SGG strain UCN34 to secrete 6 substrates via T7SSb. One of these secreted proteins, hereby named TelE, is found to be toxic when overexpressed in Escherichia coli. TelE is reminiscent of glycine zipper pore forming proteins and contains conserved glycine rich motif required for TelE toxic activity in E. coli. Toxicity of TelE was found to be partially alleviated by TipE that serves as an antitoxin. This work is of interest to people studying Streptococcus gallolyticus or T7SS as well as to the broader audience studying bacterial physiology, pathogenesis and interbacterial antagonism.

Overall, this is a well written manuscript that presents compelling data characterizing a novel type of T7SSb-secreted polymorphic pore-forming toxin. The experimental methods and approaches are appropriate and support authors' conclusions.

Comments:

Page 6, line 116: should "Type VIIb Secretion System" be abbreviated as "T7SSb" or T7bSS?

T7SSb is the correct abbreviation, to distinguish it from the original one discovered in Mycobacteria, abbreviated as T7SSa.

Page 10, line 222: I believe it should be "Fig. S5C".

We do not think so. The text correctly refers to Fig. 4D.

Page 11, line 229: Protein degradation of the TelE portion of the fusion protein could still occur?

Maybe it can occur at the amino-terminal part. However, the functional catalytic domain of TelE is located at the distal C-terminal part fused to GFP and that mutation of C-terminal Glycine at position 470 into Valine did not alter GFP fluorescence.

Page 37, line 957: In the Figure 5 legend, "(B)" is mislabeled as "(C)"

Thank you. We have changed that.

Fig 4B: Perhaps it would be advantageous to quantify the microscopy data presented in Fig. 4B.

We do not see what additional information this could bring as compared to the quantification done in Fig. 5D.

Fig 5C: Why isn't HA-tag detected in the Input (I) lanes?

The HA input signal is detected as a very intense signal on the very left part of the panel 5C and thus I am not sure to understand the question. The other input lanes correspond to untagged TeE or 6His-tagged TeE and thus HA is not detected.

June 22, 2023

Dr. Shaynoor Dramsi
Institut Pasteur
28 rue du Dr Roux
Paris
France

Re: Spectrum01481-23R1 (Characterization of TelE, a T7SS LXG effector exhibiting a conserved C-terminal glycine zipper motif required for toxicity)

Dear Dr. Shaynoor Dramsi:

Your manuscript has been accepted, and I am forwarding it to the ASM Journals Department for publication. You will be notified when your proofs are ready to be viewed.

Sincerely,

Christopher LaRock
Editor, Microbiology Spectrum
